# A study of the impact of data sharing on article citations using journal policies as a natural experiment

**Garret Christensen**[1]*, **Allan Dafoe**[2], **Edward Miguel**[3], **Don A. Moore**[3], **Andrew K. Rose**[3]

**1** U.S. Census Bureau, Washington, DC, United States of America, **2** University of Oxford, Oxford, England, United Kingdom, **3** University of California, Berkeley, California, United States of America

* garret@berkeley.edu

**Data Availability Statement:** All data files are available on the Open Science Framework: https://osf.io/pxdch/ and https://osf.io/cdt8y/.

## Abstract

This study estimates the effect of data sharing on the citations of academic articles, using journal policies as a natural experiment. We begin by examining 17 high-impact journals that have adopted the requirement that data from published articles be publicly posted. We match these 17 journals to 13 journals without policy changes and find that empirical articles published just before their change in editorial policy have citation rates with no statistically significant difference from those published shortly after the shift. We then ask whether this null result stems from poor compliance with data sharing policies, and use the data sharing policy changes as instrumental variables to examine more closely two leading journals in economics and political science with relatively strong enforcement of new data policies. We find that articles that make their data available receive 97 additional citations (estimate standard error of 34). We conclude that: a) authors who share data may be rewarded eventually with additional scholarly citations, and b) data-posting policies alone do not increase the impact of articles published in a journal unless those policies are enforced.

## Introduction

Verifiability and replicability are fundamental to science. The Royal Society's motto "*nullius in verba*" ("take nobody's word for it") encourages scientists to verify the claims of others. By sharing data, scientists can increase the verifiability and credibility of their claims. Most academic journals and professional societies encourage researchers to share their data, but these are often informal recommendations; until recently, few journals required it.

The ease of posting data on the internet has lowered the cost of data sharing; accordingly, advocates of open science have argued that data posting should be standard practice [1], and a growing number of scientific journals have started requiring that authors publicly post their data. However, this requirement remains more the exception than the rule in many fields, and researchers have not routinely posted their data unless journals require them to do so [2–4].

Researchers give several reasons for their failure to post data. Some highlight costs to the individual, including the effort required, the potential for being scooped, and the risk of being

**Funding:** GC, EM Laura and John Arnold Foundation, grant number 040951. http://www.arnoldventures.org Publication made possible in part by support from the Berkeley Research Impact Initiative (BRII) sponsored by the UC Berkeley Library. The funders had no role in study design, data collection and analysis, decision to publish, or preparation of the manuscript.

**Competing interests:** The authors have declared that no competing interests exist.

shown to be in error. But there are also benefits from posting data. If research with posted data is more persuasive or believable, it might have greater impact. Moreover, data sets are often useful for analyses that the original author(s) did not think of or chose not to conduct; researchers typically cite an article if they utilize its posted data. It seems plausible that sharing the data used in an article would increase its citations. In fact, several papers across disciplines over the last decade consistently show a positive association between data sharing and citations: in disciplines as varied as cancer microarray trials [5], gene expression microarrays [6], astrophysics [7, 8], paleoceanography [9], and peace and conflict studies [10] data sharing has been shown positively associated with citations, and computational code sharing has been shown to be positively associated with citations in the image processing literature [11]. These studies range in size (from N = 85 to N>10,000) and find a range of estimates of the increase (from a low of 9% to a high of 69% with most between 20% to 40%). They typically focus on one discipline or subject area. To our knowledge none attempt to explicitly take advantage of a change in journal policy to estimate a treatment effect.

The objective of this paper is to determine whether sharing data for a research article results in more citations for the article, using changes to journal policies as a natural experiment. If sharing data does result in increased citations, then the private benefits to data sharing could popularize the practice and improve science. We are able to study this issue across a wide variety of disciplines, enhancing the generalizability of our findings. However, we are limited since we cannot ascertain *why* publicly posted data would or would not lead to additional citations, merely *whether* articles garner more citations. We are also limited to observational rather than experimental data; we discuss the methods employed to deal with this below.

## Methods

A simple comparison of citations between articles published with and without posted data is difficult to interpret. Authors who post their data may be systematically different from those who do not, as they may choose to publish in different journals. We try to minimize this problem by focusing on journals which began, in principle, to *require* data posting. This natural experiment enables us to compare articles published before and after the change, exploiting plausibly exogenous variation in data availability caused by shifts in editorial policy.

Two separate and independent teams of researchers, both based at the University of California, Berkeley, serendipitously learned of each other's plans to exploit the natural experiment caused by shifts in journal policy. One team (Moore and Rose, hereafter MR) collected a broad sample of articles and use within-journal variation in citations to keep the journal (and thus the research community) constant, focusing attention on the effect of the change in data sharing policy. The other team (Christensen, Dafoe, and Miguel, hereafter CDM) collected deeper, more detailed data on a smaller subset of articles, without knowing the results from the MR sample. The data collection processes of both teams appear in Fig 1, and the appendix explains the timing of the researchers' interactions.

### Broad analysis

To exploit the change in journal policies, MR systematically searched the top 250 scientific journals, as identified by SCImago (http://scimagojr.com), and identified all those that changed their policies to require data posting for published articles.

Following the MR pre-analysis plan (https://osf.io/pxdch/), we collected citation count data for empirical articles–those that analyze quantitative data–published immediately *after* a change in data posting policy, as well as analogous citation counts for articles published in the period *before* the regime change. The MR analysis examined, for each journal, 200 empirical

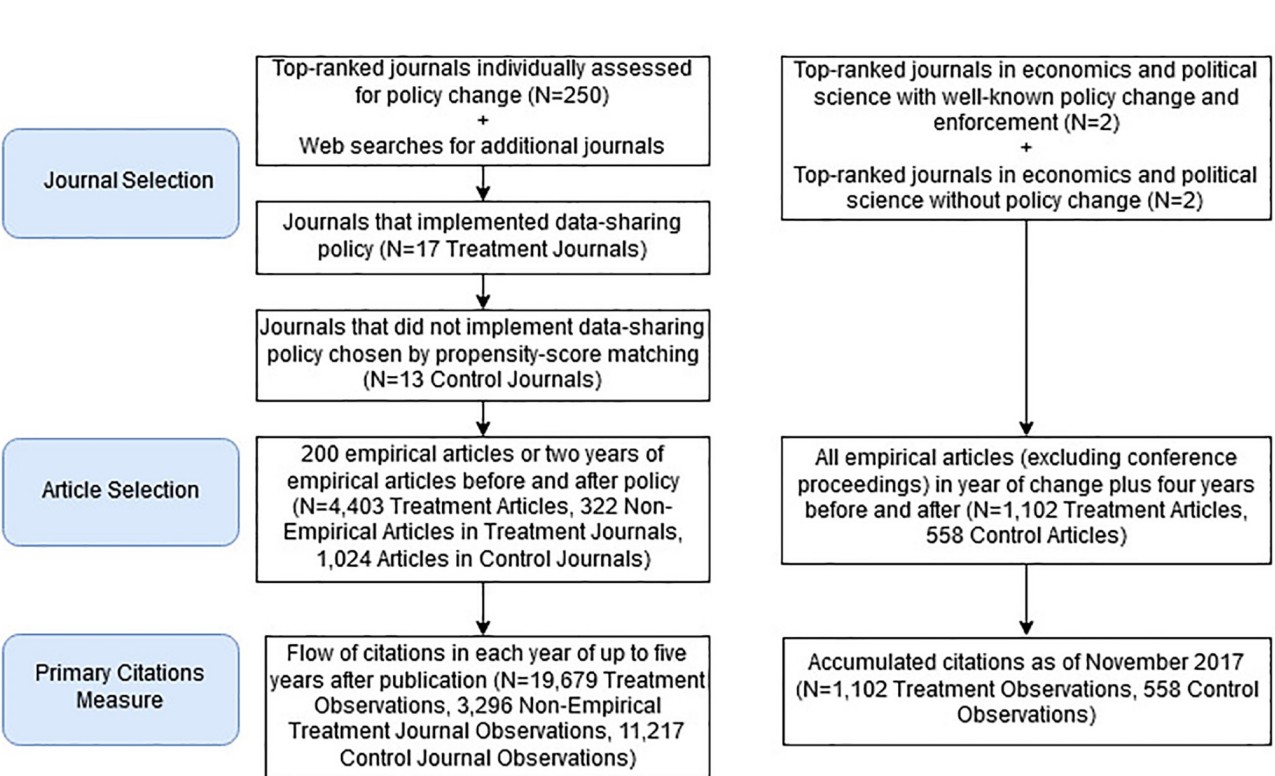

**Fig 1. Flow diagram of sample selection.** Separately our two teams selected journals, then used all empirical articles in those journals as treated observations.

articles or two years' worth of articles (whichever was less) on either side of a policy change. Research assistants recorded the annual flow of new *Web of Science* citations received one, two, three, four, and five years post-publication for each of these articles. This enables us to compare the difference in citations for articles published in the same journal shortly before and after a new data sharing requirement.

To account for the possibility of events that influenced both the change in journal policy and citation rates that could bias estimates, we collected comparable data for two natural comparison sets. We gathered data on *theoretical* articles published in the same journals; since these do not use empirical data, their citations should be largely unaffected by any change in data posting policy. We also matched the 17 *treatment journals* (which began to require data sharing) to *control journals* (which did not) and collected comparable citation data for empirical articles published in these control journals.

We selected control journals using conventional one-to-one propensity score matching with replacement [12,13] from top-ranked journals that most closely match the treatment journals on SCImago criteria. Our objective was to identify control journals that did not require data posting, but that were otherwise as similar as possible to the treatment journals in terms of observable characteristics. Accordingly, we began with non-treatment journals—that never required data posting–from the same SCImago "Top 250" list of journals where we identified our treatment journals. We matched treatment to control journals using the six indicators used to create the SCImago list itself. These criteria appear on the Scimago website. The

six variables are: a) the journal's h-index; b) the total number of citable documents published in the journal's last three years; c) citations per document over the last two years; d) references per document; e) the country where the journal was published; and f) the category of the journal. Since there are only two countries where our treatment journals were published, we created a binary variable for journals published in the UK, leaving US journals as the default. And since we only have a limited number of treatment journals, we consolidated journal category into eight areas: a) Biology; b) Ecology; c) Economics; d) Medicine; e) Molecular Biology; f) Multidisciplinary; g) Sociology and Political Science; and h) Miscellaneous.

After creating the data set, we created a binary variable, coded 1 for our treatment journals and 0 for all remaining journals (potential control journals). We then estimated a cross-sectional probit equation; the results are tabulated in our online supplement. After estimating the probit model, we then matched each of the treatment journals to a single control journal, using the closest possible journal, by predicted probit score within journal category. Sometimes this resulted in more than one treatment journal being matched to the same control journal. Both the control journals and the probit regression estimates themselves are freely available online (see https://osf.io/67c5z/). We are left with a set of 13 unique control journals along with 17 treatment journals; thus the MR analysis included data from a total of 30 distinct scientific journals. Appendix A provides the list of journals and more details on the data construction procedure.

We divide our data into citations for three types of research articles, as per above: a) empirical articles from the 17 treatment journals, our chief interest; b) theoretical articles from the treatment journals; and c) empirical articles from the 13 control journals. For each set of articles, we further split the data into articles published before and after the imposition of the data sharing policy. Control journals, by construction, do not experience any policy shift; we use the corresponding dates for the matched treatment journals.

## Deep analysis

The CDM analysis focused on two of the 17 treated journals, namely, the *American Economic Review* (*AER*) and the *American Journal of Political Science* (*AJPS*), along with comparable control journals, the *Quarterly Journal of Economics* (*QJE*) and the *American Political Science Review* (*APSR*). The CDM team pre-registered their analysis in 2015 (see https://osf.io/hv97m/).

The results that follow are consistent with our pre-registered analysis plan, with the following three exceptions: First, we did not think to exclude articles that do not use data. (Results on the full sample, described in Appendix Tables B21 to B23, are generally weaker). Second, we planned to control for time using cubic functions of months since publication. The results presented in the paper instead use year-discipline fixed effects to more flexibly control for time, as can be seen in the equations below. (Models with the cubic function yield nearly identical results). Third, in September 2017 CDM expanded their analysis from the pre-registered field of political science (from which they had seen results) to include the economics portion of the project, which had been previously discussed, but not pre-registered; this was before seeing any results from the broad analysis conducted by MR.

Fig 1 shows the sample selection procedure. We chose *AJPS* and *AER* as treatment journals because of their prominence in their fields, and because they are known for relatively strong enforcement of data-posting requirements. We examined each article published in these four journals around the time of the change in journal policy in order to determine the availability of data (as well as the code) required to reproduce the article's results.

Specifically, CDM collected data on articles published in the *AER* from 2001 to 2009, centering +/- four years around the 2005 change in data policy; articles from the *QJE* (which had

no comparable policy change during this period) serve as a comparison group. We collected data from *AJPS* articles from 2006 through 2014, since the journal experienced enhanced data posting policy changes in both 2010 and 2012; *APSR* serves as the control journal to *AJPS* for identifying potentially confounding temporal effects.

CDM regressed total accumulated citations from Elsevier's Scopus database as of November 2017 (unlike the MR analysis which used flow of citations) on article characteristics and journal policy using ordinary least squares linear regression (OLS). Because it remains possible that articles sharing data are different from those that do not in other dimensions, thus biasing a simple comparison, we also attempt to identify and control for other plausible predictors. We classified articles by research method (specifically, experimental, observational, and theoretical), by subject matter, and by the top institutional ranking of the authors (online Appendix B provides details on the data construction). Observations with missing data are excluded.

The OLS estimates are still liable to suffer from omitted variable bias (for example, if article quality is correlated with both data sharing and citations). To address this potential bias, CDM also use the changes in journal policies as instrumental variables for data availability, and examine the relationship between (instrumented) data availability and citations using two stage least squares (2SLS) which produces estimates which are biased but consistent [14,15]. The 2SLS model relies on two major assumptions: relevance (the data sharing policy is a strong predictor of data sharing) and the exclusion restriction (the data sharing policy only affects citations through data sharing, and not through any other channel). We explicitly test the relevance assumption and show evidence regarding the second below. We focus on models run with only those articles that use data, since articles without data are not subject to the policy. Estimation uses 2SLS regressions, per the equations below:

$$
\begin{aligned}
availability_i \;=\; & \alpha_1 + \beta_1 AER_i + \beta_2 AJPS_i + \beta_3 APSR_i + \beta_4 Post2005_i + \beta_5 Post2010_i \\
& + \beta_6 Post2012_i + \beta_7 AER * Post2005_i + \beta_8 AJPS * Post2010_i + \beta_9 AJPS \\
& * Post2012_i + \sum_d \sum_t \gamma_{td} + v_i
\end{aligned}
\tag{1}
$$

$$
\begin{aligned}
citations_i \;=\; & \alpha_2 + \eta_1 AER_i + \eta_2 AJPS_i + \eta_3 APSR_i + \eta_4 Post2005_i + \eta_5 Post2010_i \\
& + \eta_6 Post2012_i + \eta_7 \widehat{availability}_i + \sum_d \sum_t \gamma_{td} + u_i
\end{aligned}
\tag{2}
$$

Eq 1 refers to the first stage, which predicts availability of data based on journal and publication date, while Eq 2 uses predicted data availability to attempt to estimate the impact on number of total citations under assumptions we discuss below. We control for the journal (coefficient estimates $\beta_1, \beta_2, \beta_3$), date of publication (whether it is after the policy: $\beta_4, \beta_5, \beta_6$), and our instrumental variables are the interaction terms (with coefficient estimates $\beta_7, \beta_8, \beta_9$). We also flexibly account for time since publication (and the fact that total citations mechanically increase over time) with fixed effects for each discipline-year combination ($\gamma_{td}$).

To test the key assumption of this instrumental variables model (referred to as the exclusion restriction), we examine the observable characteristics of articles and assess whether these were affected by the editorial changes. If not, we can be more confident that unobservable characteristics are also largely uncorrelated with editorial changes. Accordingly, we classified articles by research method (experimental, observational, theoretical), by subject matter (American politics, public policy, international relations, comparative politics, political methodology, and political theory for political science, and *Journal of Economic Literature* topic codes for economics), and by the top institutional ranking of authors. We then employ the specification in Eq 1, but with these variables as the outcome.

## Results

### Broad analysis

Fig 2 presents six event studies using the data from the MR sample, two for each of our three combinations of articles/journals. First, consider Panel A of Fig 2, which portrays the average citations for the 4,403 *empirical* articles published in our *treatment* journals. At the left of the graph, the mean flow of new citations ($\approx$4.1) garnered in the first year after publication, for articles published *before* the policy change, is portrayed with a thin line along with a corresponding (+/- 2 standard deviation or 95%) confidence interval. Average annual citations rise with time; the right side of each panel shows new citations garnered in the fifth year after publication (again, only for articles published before the regime change). Panel B depicts the analogue for the 322 *theory* articles published in *treatment* journals, while Panel C on the right portrays that for the 1,024 *empirical* articles in *control* journals.

Panel D presents the difference between average citations for empirical articles published *before* and *after* the regime change in the treatment journals, along with the associated 95% confidence interval. Our first main finding is that empirical articles published in the 17 treatment journals just before required data sharing receive citations comparable to those published just afterwards.

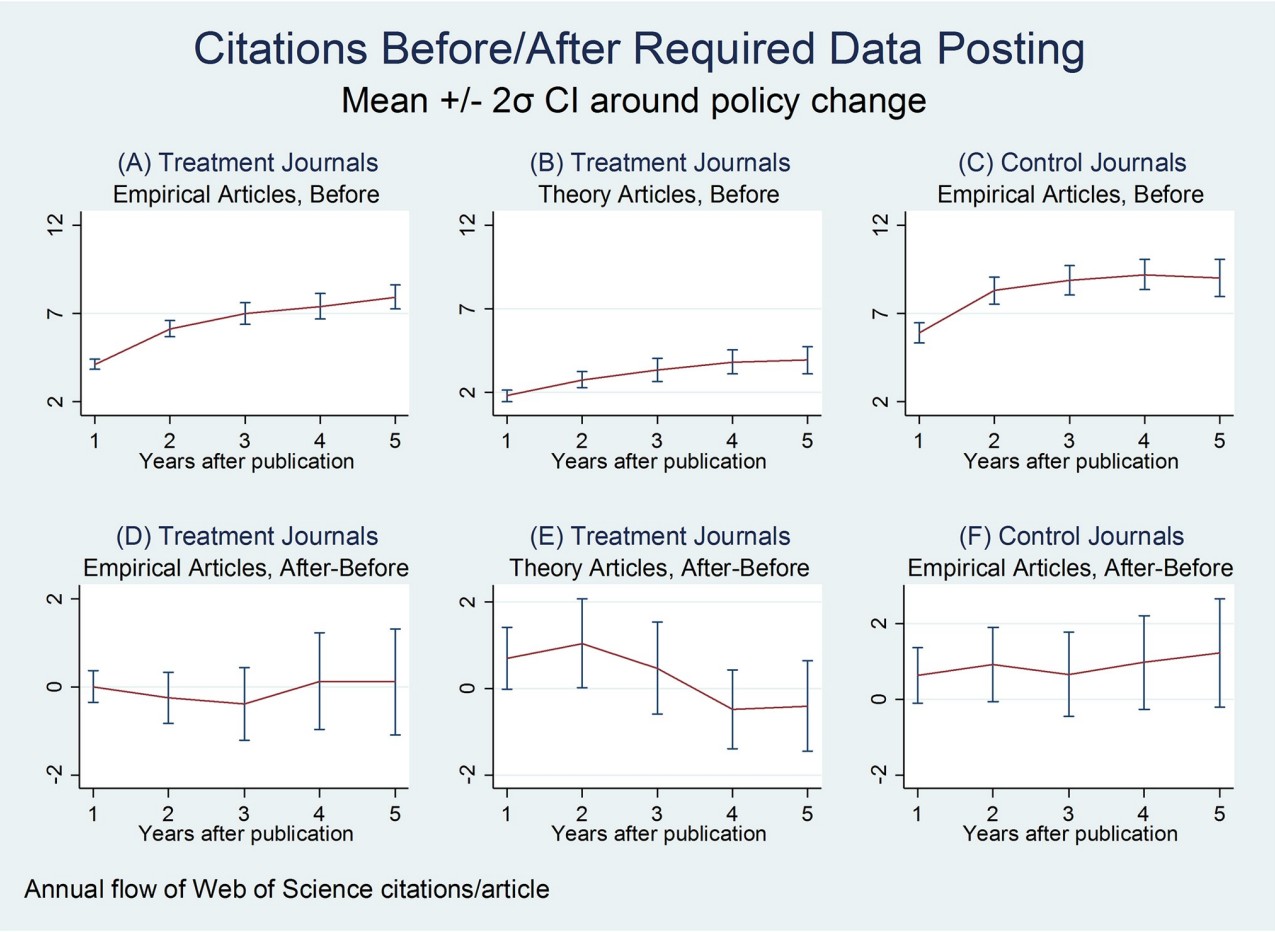

**Fig 2. Citations per article appear in the top row, for articles published before treated journals' adoption of a data-posting policy.** The bottom row shows the difference in citations per article following the policy change. Error bars show 95% confidence intervals.

**Table 1. Effect of policy switch, treating years after publication simultaneously.**

| Years since Publication | Empirical Papers, Treatment Journals | | | Theoretical Papers, Treatment Journals | | | Empirical Papers, Control Journals | | |
|---|---|---|---|---|---|---|---|---|---|
| Estimator | LS | LS | Poisson | LS | LS | Poisson | LS | LS | Poisson |
| Log(1+Cites) | N | Y | N | N | Y | N | N | Y | N |
| 1–5 | -.20 (.42) | -.03 (.05) | -.02 (.06) | .11 (.88) | -.04 (.11) | -.04 (.20) | -1.98 (2.15) | -.06 (.08) | -.21 (.16) |
| 1 | -.08 (.59) | -.00 (.07) | -.00 (.10) | .09 (.98) | -.01 (.14) | .03 (.22) | -2.14 (2.51) | -.05 (.08) | -.20 (.20) |
| 2 | -.17 (.23) | -.03 (.03) | -.02 (.03) | .18 (.48) | -.02 (.05) | .00 (.10) | -.93 (1.00) | -.03 (.04) | -.10 (.07) |
| 3 | -.14 (.16) | -.02 (.02) | -.02 (.02) | .10 (.27) | -.00 (.03) | -.01 (.06) | -.64 (.68) | -.02 (.02) | -.07 (.05) |
| 4 | .04 (.14) | .01 (.01) | .01 (.02) | .01 (.22) | -.01 (.03) | -.01 (.05) | -.45 (.54) | -.01 (.02) | -.05 (.04) |
| 5 | -.06 (.10) | -.01 (.01) | -.01 (.01) | -.06 (.17) | -.02 (.02) | -.03 (.04) | -.43 (.41) | -.02 (.02) | -.05 (.03) |

The table presents coefficients for a dummy variable (1 for after data posting required, 0 otherwise) estimated with least squares (LS) or Poisson; each column represents a separate regression. Robust standard error (clustered by journal) recorded in parentheses. Regression is annual citations/article. Controls included but not recorded: intercepts for each year elapsed since publication, fixed effects for journal and publication year, and log number of co-authors.

The divergence between average article citations before and after the policy shift is small, and is neither substantively nor statistically significant. These small differences characterize the main sample of empirical articles in the treatment journals (Panel D) as well as for the other two comparison sets of articles (Panels E and F), where null effects are expected (as placebo checks). Most strikingly, as Table 1 shows, there is no evidence that data sharing policies *raise* citations counts. However, this may be because such policies are not enforced; the existence of a data sharing policy does not necessarily require that data is actually shared.

## Deep analysis

We checked the availability of data and code for each of the 1660 empirical articles published in regular issues (i.e., excluding conference proceedings, anniversary retrospectives, and the like) of the four CDM journals. We find that data are posted for less than a quarter of articles published prior to the policy changes; this increases to approximately 70 percent for articles published immediately after the policy changes at *AER* and *AJPS*, as shown in Panels A and B respectively of Fig 3. Both journals experienced abrupt increases in data availability following the policy change, and it is this variation that we exploit in the analysis below. We analyzed sharing of both data and the statistical code necessary to replicate the analysis, as well as sharing of at least data; rates are similar for both, as are impacts on citations; for simplicity, we focus on data sharing alone (for more information, see Appendix B).

Our second main finding, based on the CDM analysis, is that articles published *after* the data posting policy change in these two journals enjoy approximately 40 percent higher total citations than those published *before*, when using total accumulated citations after a relatively long time period (as of November 2017). This is visible in panels C and D of Fig 3, which present the average difference in total citations between the treatment and control journals for articles published in a given year, and the associated confidence intervals. Differences are regularized to zero in the year before the policy was first implemented in each treatment journal. The apparent trend in the pre-treatment period in economics is a potential concern. Some of the variation in the AER-QJE pre-treatment trend can be explained by a single paper [16], published in the AER in 2001, which has been cited two orders of magnitude more than most AER articles. We also confirm that citations in years one through five after article publication do not increase in the treatment journals (see Fig 3 panels E and F), mirroring the findings in Fig 2. The delay in the effect seems reasonable given that economics and political science articles

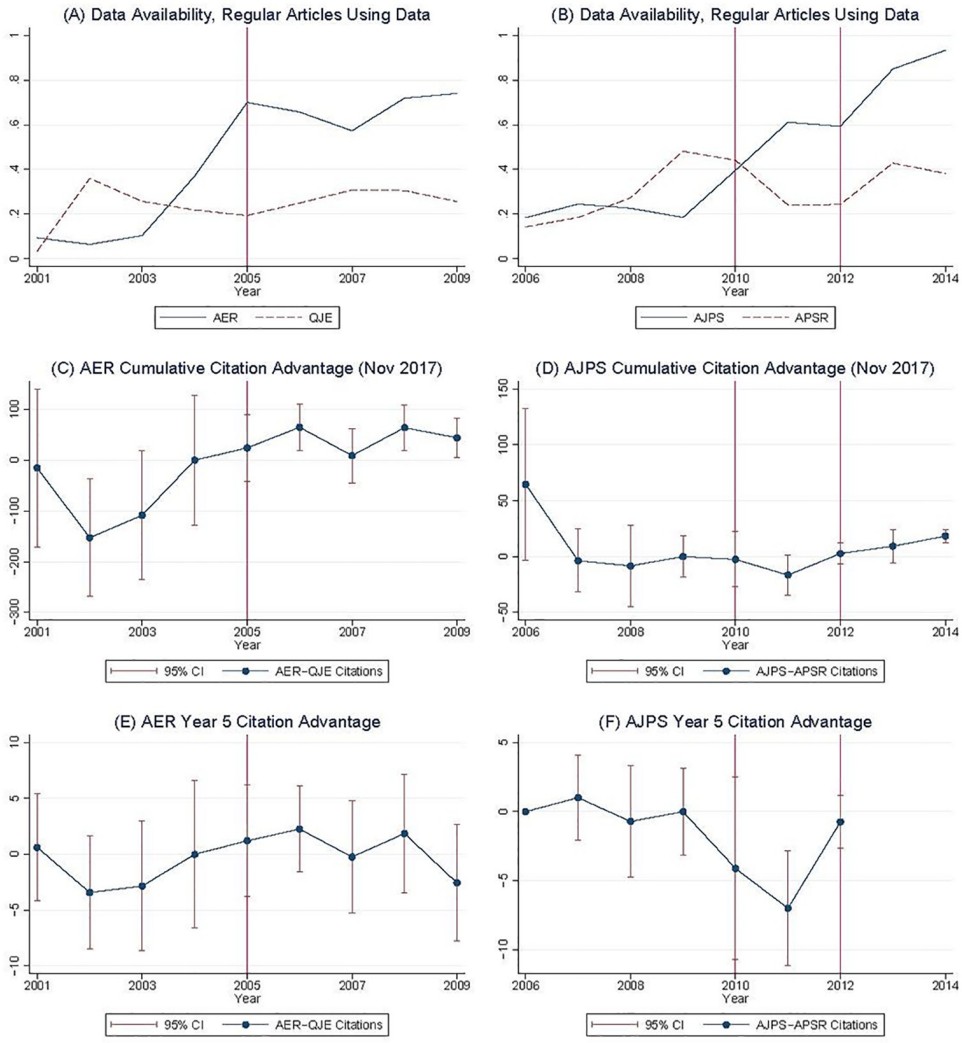

**Fig 3. Percentage of articles with posted data appear in the top row.** The middle row shows the cumulative citation advantage for articles with posted data as of November 2017, and the bottom row shows citations in year five post-publication. AER = American Economic Review, QJE = Quarterly Journal of Economics, AJPS = American Journal of Political Science, and APSR = American Political Science Review.

receive less than half of their 15-year citation total in the first five years after publication [17], and in what follows we thus focus on Panels C and D.

The findings of Fig 3 indicate a potentially positive association between journal data posting policies and subsequent article citations. Our regression analysis (both OLS and 2SLS) leads to similar conclusions. CDM collected total citations for all articles in these four journals in November 2017. The time horizon varies from three to sixteen years, since articles are published between 2001 and 2014. The main OLS regression is shown in Table 2, with strong positive associations between data sharing and citations, after controlling for journal, and year-discipline fixed effects as described above. Note that in this table, the sample descriptions for each column listed as "Data-Only" imply using only articles that use data. "Data-NoPP" implies a further limitation of removing the *Papers & Proceedings* conference and similar anniversary-style issues. "Data-Econ" or "Data-NoPP-Econ" imply limiting to only data articles from economics. For the specifications with both disciplines in the first and second columns,

**Table 2. OLS estimates of citations and data sharing.**

| VARIABLES | (1) Citations | (2) Citations | (3) Citations | (4) Citations |
|---|---|---|---|---|
| Data Available | 64.15*** | 52.96*** | 93.47*** | 87.23*** |
| | (7.61) | (9.38) | (11.42) | (16.75) |
| AER | -130.05*** | -100.36*** | -130.59*** | -107.77*** |
| | (10) | (12.08) | (11.93) | (15.96) |
| AJPS | -189.75*** | -160.18*** | | |
| | (21.59) | (25.36) | | |
| APSR | -166.55*** | -138.07*** | | |
| | (22.31) | (26.1) | | |
| Observations | 2,210 | 1,660 | 1,467 | 920 |
| R-squared | 0.18 | 0.2 | 0.15 | 0.12 |
| Year-Discipline FE | Yes | Yes | Yes | Yes |
| Sample | Data-Only | Data-NoPP | Data-Econ | Data-NoPP-Econ |
| Mean Dep. Var. | 99.03 | 116.1 | 122.8 | 167.4 |

Standard errors in parentheses

*** p<0.01,

** p<0.05,

* p<0.1

Table features heteroskedasticity-robust linear regressions of article citations on whether an article has shared data. All regressions included year-discipline fixed effects. The sample for each column is defined as follows: "Data-Only" uses only articles that use data. "Data-NoPP" further removes the *Papers & Proceedings* conference and similar anniversary-style issues. "Data-Econ" or "Data-NoPP-Econ" limits to only data articles from economics.

the coefficient for data sharing ranges from 52.96 (standard error 9.38) to 64.15 (standard error 7.61) additional citations.

Results from the 2SLS regression also show large positive effects from data sharing, as shown in Table 3. F-statistics from the first stage of these regressions are over 20 in the specifications that include both disciplines. The table shows published articles with posted data enjoyed an increase of 97.04 (standard error 34.12) to 109.28 (standard error 41.15) total citations over a mean of approximately 100, suggesting nearly a doubling.

However, when we test the exclusion restriction assumption of our 2SLS model, the timing of the change in editorial policy regarding data posting does appear to have been associated with a modest change in the types of articles published in the *AER* (relative to the *QJE*). Experimental articles increased by 9 percentage points and observational empirical studies decreased by 12 percentage points around the time of the policy change; both estimates have standard errors of 5 percentage points and are thus significant at the 10 percent and 5 percent significance level, respectively. Submissions from the most prestigious institutions increased relative to those from only slightly less prestigious universities by as much as 24 percentage points, statistically significant at the one percent level. We observe no such changes in article or author characteristics at *AJPS* at the time of its data policy change.

## Discussion

Our main results seem to indicate that for most journals, data sharing policies do not lead to increased citations in the five years following publication. The same is true of theory articles published in the treatment journals, and empirical articles in the control journals, as shown in Panels E and F of Fig 1 respectively. However, sharing data does appear to lead to increased citations over a longer time period at the *American Economic Review*, as shown in Tables 2 and 3. Given the results of our tests of the exclusion restriction in Table 4, however, we cannot

**Table 3. 2SLS estimate of data sharing and citations.**

| VARIABLES | (1) Citations | (2) Citations | (3) Citations | (4) Citations |
|---|---|---|---|---|
| Data Available | 97.04*** | 109.28*** | 672.53*** | 234.55*** |
| | (34.12) | (41.15) | (259.61) | (86.28) |
| AER | -130.33*** | -112.57*** | -142.14*** | -139.67*** |
| | (15.91) | (23.59) | (23.51) | (32.07) |
| AJPS | -204.19*** | -181.21*** | | |
| | (26.1) | (26.71) | | |
| APSR | -173.91*** | -147.74*** | | |
| | (20.57) | (20.72) | | |
| Post-Mar 2005 | -24.78 | -16.52 | 72.03 | -28.14 |
| | (29.22) | (32.28) | (78.05) | (36.52) |
| Post-Oct 2010 | -32.34 | -35.64 | | |
| | (22.12) | (24.36) | | |
| Post-July 2012 | -8.54 | -8.86 | | |
| | (10.19) | (11.31) | | |
| Observations | 2,210 | 1,660 | 1,467 | 920 |
| R-squared | 0.17 | 0.18 | -1.34 | 0.04 |
| Year-Discipline FE | Yes | Yes | Yes | Yes |
| Sample | Data-Only | Data-NoPP | Data-Econ | Data-NoPP-Econ |
| Mean Dep. Var. | 99.03 | 116.1 | 122.8 | 167.4 |
| F Stat | 20.14 | 32.94 | 9.738 | 48.03 |

Robust standard errors in parentheses

*** p<0.01,

** p<0.05,

* p<0.1

Table features 2SLS regressions of article citations on whether an article has shared data, with journal policy requirements as an instrument for data sharing. All regressions included year-discipline fixed effects. The sample for each column is defined as follows: "Data-Only" uses only articles that use data. "Data-NoPP" further removes the *Papers & Proceedings* conference and similar anniversary-style issues. "Data-Econ" or "Data-NoPP-Econ" limits to only data articles from economics and drops political science.

completely rule out the possibility that changes in the types of authors or articles contributed to higher rates of data sharing following the new data policies at the *AER*.

What can explain the null results for most journals? One obvious concern is the consistency with which journals actually enforced data-posting policies. There is ample evidence that journal data posting policies are enforced weakly, if at all. We found that rates of data availability for empirical articles published after journals adopted data-sharing policies differ widely between journals, from 0% to 83%, with a mean of 35% (see Table A17 of supplemental online materials). Examining the effect of actual data availability at the level of the individual article clarified the results. As shown by the increase in the fraction of articles that share data (Panels A and B in Fig 3), the policies seem to be at least partially enforced for these journals: the first stage in the 2SLS regression is strong, with F-statistics generally over 20 (Table 2). Confirming the presence of posted data is difficult, requiring systematic internet searches across many possible locations. However, it appears that journals with strictly enforced policies may see increased citations.

One limitation of our study is that the analysis does not reveal precisely why publicly posting data might help an article garner citations. Future research could examine whether shared data is an indicator of the reliability of results, or an input that is used in subsequent research. Still, regardless of the underlying mechanisms, sharing data represents the provision of a public good that benefits the scientific enterprise. It allows scientists to enact the Royal Society's

**Table 4. Tests of the 2SLS exclusion restriction.**

| VARIABLES | (1) Experimental | (2) Observational | (3) Top 1 | (4) Top 10 | (5) Top 20 | (6) Top 50 |
|---|---|---|---|---|---|---|
| AER post-2005 Policy | 0.09* | -0.12** | 0.24*** | -0.13*** | -0.06 | -0.01 |
| | (0.05) | (0.05) | (0.06) | (0.05) | (0.05) | (0.05) |
| AJPS post-2010 Policy | 0.13* | -0.12 | 0.03 | 0.05 | 0.12 | 0.07 |
| | (0.07) | (0.08) | (0.09) | (0.07) | (0.08) | (0.08) |
| AJPS post-2012 Policy | -0.13* | 0.11 | -0.04 | 0.03 | -0.02 | 0.06 |
| | (0.08) | (0.08) | (0.09) | (0.07) | (0.08) | (0.09) |
| Observations | 1,658 | 1,658 | 1,653 | 1,653 | 1,653 | 1,653 |
| R-squared | 0.02 | 0.03 | 0.11 | 0.03 | 0.01 | 0.09 |
| Year-Discipline FE | Yes | Yes | Yes | Yes | Yes | Yes |
| Sample | Data-NoPP | Data-NoPP | Data-NoPP | Data-NoPP | Data-NoPP | Data-NoPP |
| Mean Dep. Var. | 0.139 | 0.842 | 0.266 | 0.117 | 0.155 | 0.195 |

Standard errors in parentheses

*** $p < 0.01$,

** $p < 0.05$,

* $p < 0.1$

Table features OLS regressions of article characteristics on whether the article was in a journal after a data sharing policy requirement was enacted. The sample in all specifications (described as "Data-NoPP") uses only articles with data and further removes the *Papers & Proceedings* conference and similar anniversary-style issues.

motto and examine evidence for themselves, moving closer to the ideal of open and collaborative scientific inquiry [18].

The two main results, taken together, indicate that it is not sufficient for scientific journals merely to *announce* a data sharing requirement. This appears generalizable across a wide group of disciplines, given the range of fields covered by the MR analysis. Without diligent enforcement, a toothless journal data policy appears to produce the same result as no policy at all; few authors post their data. But even if *de jure* journal policy does not guarantee data sharing, our results indicate that public data sharing can eventually yield private benefits for scholars, in the form of enhanced citations, which provide meaningful *de facto* incentives to share scientific data.

## Supporting information

**S1 STROBE Checklist. STROBE statement—Checklist of items that should be included in reports of observational studies.**
(DOCX)

**S1 File.**
(DOCX)

## Acknowledgments

The authors thank the editor Florian Naudet, and two referees for useful suggestions. Rose thanks the Central Bank of Barbados and the National University of Singapore for hospitality during the course of this research. Any opinions and conclusions expressed herein are those of the authors and do not necessarily reflect the views of the U.S. Census Bureau.

## Author Contributions

**Conceptualization:** Allan Dafoe, Edward Miguel, Don A. Moore, Andrew K. Rose.

**Data curation:** Garret Christensen, Edward Miguel, Don A. Moore.

**Formal analysis:** Garret Christensen, Allan Dafoe, Edward Miguel, Don A. Moore, Andrew K. Rose.

**Funding acquisition:** Garret Christensen, Edward Miguel.

**Investigation:** Garret Christensen, Don A. Moore, Andrew K. Rose.

**Methodology:** Garret Christensen, Edward Miguel, Don A. Moore, Andrew K. Rose.

**Project administration:** Garret Christensen, Edward Miguel, Don A. Moore.

**Resources:** Don A. Moore.

**Software:** Garret Christensen.

**Supervision:** Garret Christensen, Don A. Moore.

**Validation:** Garret Christensen, Andrew K. Rose.

**Visualization:** Garret Christensen, Andrew K. Rose.

**Writing – original draft:** Garret Christensen, Don A. Moore, Andrew K. Rose.

**Writing – review & editing:** Garret Christensen, Allan Dafoe, Edward Miguel, Don A. Moore, Andrew K. Rose.

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
