## [Decision Letter · Decision Letter 0]

26 Jul 2019

PONE-D-19-18102

Does Data Sharing Increase Citations?

PLOS ONE

Dear Dr. Christensen,

Thank you for submitting your manuscript to PLOS ONE. After careful consideration, we feel that it has merit but does not fully meet PLOS ONE’s publication criteria as it currently stands. Therefore, we invite you to submit a revised version of the manuscript that addresses the points raised during the review process.

**I would like to thank here the two reviewers who assessed your manuscript as fast as possible.**

One reviewer was somewhat enthusiast and asked for major revisions. He made very precise comments. **Your answers to these comments will be all the more important.**

The second reviewer recommended to reject the paper because the overall presentation of your manuscript was inadequate and unclear. He asked for an extensive revision following current reporting guidelines (please select the more appropriate for your paper from guidelines from the EQUATOR network). And I must say that I agree with him. He proposed to reject the paper and suggested a resubmission of an extensively amended version. I rather think that asking for “major revisions” would be a better compromise. Reporting research at the highest standards is an important aspect of research reproducibility and this point should be consequently improved before any decision on your paper. I recognize that there are no perfect guidelines for your study design but some may be relevant (PRISMA for a flow diagram of identified papers / STROBE for the handling of observational data etc.). However, you must be informed that a decision of major revision does not imply any commitment to accept your manuscript unless your alterations meet the standards of the review. In the present manuscript, revisions are expected to be extensive.

We would appreciate receiving your revised manuscript by Sep 08 2019 11:59PM. To enhance the reproducibility of your results, we recommend that if applicable you deposit your laboratory protocols in protocols.io, where a protocol can be assigned its own identifier (DOI) such that it can be cited independently in the future. For instructions see: http://journals.plos.org/plosone/s/submission-guidelines#loc-laboratory-protocols

We look forward to receiving your revised manuscript.

Kind regards,

Florian Naudet, M.D., M.P.H., Ph.D.

Academic Editor

PLOS ONE

Journal Requirements:

2) Please include captions for your Supporting Information files at the end of your manuscript, and update any in-text citations to match accordingly. Please see our Supporting Information guidelines for more information: http://journals.plos.org/plosone/s/supporting-information.

Reviewers' comments:

Reviewer's Responses to Questions

**Comments to the Author**

1. Is the manuscript technically sound, and do the data support the conclusions?

Reviewer #1: Yes

Reviewer #2: Partly

2. Has the statistical analysis been performed appropriately and rigorously? 

Reviewer #1: Yes

Reviewer #2: I Don't Know

3. Have the authors made all data underlying the findings in their manuscript fully available?

Reviewer #1: Yes

Reviewer #2: Yes

4. Is the manuscript presented in an intelligible fashion and written in standard English?

Reviewer #1: Yes

Reviewer #2: No

5. Review Comments to the Author

Reviewer #1: This is a carefully planned and performed study dedicated to the impact of data sharing on citations, which should be published. Looking at three types of research articles (empirical papers from 17 treatment journals), theoretical papers from treatment journals and empirical papers from 13 control journals, the first part of the analysis shows that empirical papers published in the 17 treatment journals just before required data sharing receive citations comparable to those published afterwards. The results are explained by the inconsistency with which journals actually enforced data posting policies. Because data availability is difficult to assess, analysis was focussed on 2 treatment journals in part 2 of the study. Here again, citations in year 1 through 5 after article publication do not increase in the treatment journals. With the argument of delay of citations for economics and political science articles, the authors then focus on total citations accumulated as of November 2017, covering a time horizon between 3 and 16 years. The authors found, using 2 stage least squares regression, that there was nearly a doubling of citations. In order to interpret this positive result correctly, several questions should be answered:

Pre-analysis plan

a) In the data sharing and citations pre-analysis plan, which was published in 2015, data are used from all articles from 2006 through 2014 in AJPS and from APSR as control group. In the second part of the manuscript the authors focused instead on 2 treated journals (AER, AJPS) along with comparable control journals (QJE, APSR), The authors should explain why this extension of focus was performed.

b) Table A14 in the supplement presents individual results when β is estimated using data on empirical papers from a single journal. Only 2 journals (AER, JLE) deliver consistently positive estimates of beta at conventional significance levels. AER was selected for the second part of the study. Could it be that the decision for having a look at AER was made after this first analysis was performed?

c) In the first part of the study citations 1-5 years after publication are considered and in the second part 5-year citation advantage both with negative statistical results. In part 2, the analysis is extended by using total citations accumulated at November 2017, which may cover longer periods. The reviewer could not find information whether the analysis period was specified in advance or not (1-5 years following publication or total citations accumulated at November 2017).

In summary, it is not totally clear to the reviewer, whether the 2nd part of the study followed the pre-analysis plan and if not, where it deviates. This may have influence on the interpretation of statistical significance (alpha error).

Regression model and bias

Citations in year 1 through 5 after article publication do not increase in the treatment journals (AER, AJP); see figure 2, panel E, F). If the cumulative citation advantage (Nov. 2017) is considered, the 2 journals enjoy approximately 40% higher total citations (see figure 2, panel C,D).

a) In figure 2, panel D, the 95% CI is extremely small for 2014 and AJPS Cumulative Citation Advantage (Nov 2017). Apart from this value, the AJPS cumulative citation advantage seems to be small. The authors should explain this.

b) The apparent trend in the pre-treatment period in economics is a potential concern for the authors and explained by a highly cited AER paper. The reviewer does not understand this argument because AER – QJE is negative in the pre-treatment period.

c) From the 2-stage least squares regression it is concluded that published articles with posted data enjoyed an increase of 90 to 125 citations over a mean of approximately 100, suggesting nearly a doubling of citations. Here the question is how good the goodness-of-fit of the model is. The R-squared values are not very impressive. Residual plots to determine whether the coefficient estimates are biased could not be found in the supplementary material. The authors should demonstrate the adequacy of the model fit.

Reviewer #2: Dear authors,

below you can find comments that might improve your manuscript in different perspectives. Please enface my comments as a way to improve your piece of research and I will be more than happy to re-appraise it in a more suitable version.

a) This is a methodological paper and reporting guidelines for methodological research is under discussion. However, you may borrow some items (or even in an intuitive manner) from reporting guidelines from other designs to better guide your writing. Your manuscript in the current form is poorly written and unidentifiable, what is even more crucial for a paper that deals with a question related, at least in some level, to science reproducibility. For example, a simple item like the title does is flawed - it does not indicate if is it an original experiment, a letter, an editorial etc. Standards of reproducible reporting should be addressed (please check www.equator-network.org) and some items might be included accordingly to allow the reproducibility of your paper - title, structured abstract, primary and secondary outcomes, eligibility criteria, search strategy (or the deep description of the approach to address conveniently the journals) and etc. You've also waste too much words with non-informative sentences. Please go straight to the point. Avoid also the use of rhetorical sentences and the excess of latim expressions (although suitable at some points).

b) Structure your manuscript. Methods, Results and Discussion are completely unorganized. I don't care about the inversion of results and discussion in terms of the length (so, the fact of your discussion was tiny was not a problem for me), if the journal permits. However, they should be informative and address the point to why they were created originally - and you've left a lot of discussion informations in the results section. As for the methods part, this is critical. I must confess I went lost several times when reading your methods due to the way you presented this. It does not favour you neither the reader. Please think careful about it. We need to acknowledge sharply the waste in research nowadays and I don't think your research is waste.

c) In your methods, I'm very concerned in the way you've designed your experiment. Propensity scores may balance non-randomized studies at baseline, however, caveats are widely known. Beyond the fact the description of your control group and your propensity score model (I don't know for what variables you've adjusted) are poorly written, I'm hesitant if it is the best method to match groups for your experiments, if is it possible. Given the data on citations related to data-sharing is very naive, I don't know if I would rather prefer to first describe the rate of citations instead of making a comparison. Also, the rate of citation for papers, doesn't matter the data-sharing regimen, is usually low within the first years when there is a salient amount of citations; and, if linear, we would have a decent number only after years. Therefore, you may incur in a comparison with a low frequency of events (with or without inferential analysis) and thus I re-affirm I'm not convinced this is the time to compare citations, but yes for a description. In a future, I would say a comparison within the same journal might be more appropriate, once one will always need to deal with confounding factors that would be difficult to input in a model like the merit of a paper (which makes it more cited), publication bias, gender bias, geographic bias and others. As a last mention, please be very detailed with your control group. I couldn't follow it.

d) The rationale seems weak to me. Do you really think the perception of integrity might improve citations, at least nowadays? I mention the "thinking" because we don't have empirical evidence to support this. I know for a given topic when still under investigated, we have little evidence to support our kick-offs, but I don't know if is it the critical point, or even if data-sharing might incur in the augment of citations (unless for the use of the data, as you acknowledged in your paper). Citations are a multidimensional approach and a multivariable-explained phenomenon. Anecdotally, I'm not convinced data-sharing may play a role in citations within the myriad of variables - if researchers nowadays didn't perceive the value of data-sharing in various scenarios, I would say that an association is still harder (although not implausible by nature). Even, as a data-sharing researcher, I think data sharing should be observed as a way to reduce the waste in research; to improve the reproducibility and verifiability of a given finding; to augment the utility of a finding; and an ethical obligation of every single researcher regardless of the experimental addressed designed - what you brilliantly did. In time, congratulations by the way you shared the data. I didn't re-analyze your findings, but it is clear your compliance with reproducibility standards in that way.

6. PLOS authors have the option to publish the peer review history of their article (what does this mean?). If published, this will include your full peer review and any attached files.

Reviewer #1: No

Reviewer #2: Yes: Lucas Helal

---

## [Author Response · Author response to Decision Letter 0]

31 Aug 2019

Reviewers,

We thank you for your careful review and provide specific responses below. Reviewer comments are repeated with our responses below. This information is also included (with formatting) in our uploaded response letter to the editor.

Reviewer #1: This is a carefully planned and performed study dedicated to the impact of data sharing on citations, which should be published. Looking at three types of research articles (empirical papers from 17 treatment journals), theoretical papers from treatment journals and empirical papers from 13 control journals, the first part of the analysis shows that empirical papers published in the 17 treatment journals just before required data sharing receive citations comparable to those published afterwards. The results are explained by the inconsistency with which journals actually enforced data posting policies. Because data availability is difficult to assess, analysis was focussed on 2 treatment journals in part 2 of the study. Here again, citations in year 1 through 5 after article publication do not increase in the treatment journals. With the argument of delay of citations for economics and political science articles, the authors then focus on total citations accumulated as of November 2017, covering a time horizon between 3 and 16 years. The authors found, using 2 stage least squares regression, that there was nearly a doubling of citations. In order to interpret this positive result correctly, several questions should be answered:

Pre-analysis plan

a) In the data sharing and citations pre-analysis plan, which was published in 2015, data are used from all articles from 2006 through 2014 in AJPS and from APSR as control group. In the second part of the manuscript the authors focused instead on 2 treated journals (AER, AJPS) along with comparable control journals (QJE, APSR), The authors should explain why this extension of focus was performed.

There was internal debate from the beginning amongst the authors involved in this piece of the project (Christensen, Dafoe, Miguel) about whether to add the additional journals (AER and QJE). The desire to collect the AER/QJE data was always present, but it was initially unclear how feasible the data gathering process would be, especially because it was Christensen’s first time managing a team of undergraduate researchers. The AER/QJE data was not gathered until after authors viewed preliminary results from the AJPS/APSR sample. Conservative readers should be aware that the addition of the AER/QJE sample was not pre-specified. We now explicitly mention this point in the revised version of the paper.

b) Table A14 in the supplement presents individual results when β is estimated using data on empirical papers from a single journal. Only 2 journals (AER, JLE) deliver consistently positive estimates of beta at conventional significance levels. AER was selected for the second part of the study. Could it be that the decision for having a look at AER was made after this first analysis was performed?

This portion of the analysis was conducted independently by separate teams (Moore, Rose responsible for A14; Christensen, Dafoe, Miguel, responsible for adding the AER/QJE sample after having seen AJPS/APSR results, but before having seen any results of any sort from the Moore, Rose team.) We thoroughly agree that the timing of our studies is both sub-optimal and confusing to the reader, but the research projects truly did stumble upon each other serendipitously mid-stream, so a perfect combination was impossible. We hope that we are now fully transparent on this issue, and that our re-written Introduction and Methods sections clearly explain this.

c) In the first part of the study citations 1-5 years after publication are considered and in the second part 5-year citation advantage both with negative statistical results. In part 2, the analysis is extended by using total citations accumulated at November 2017, which may cover longer periods. The reviewer could not find information whether the analysis period was specified in advance or not (1-5 years following publication or total citations accumulated at November 2017).

As described above, the two separate teams began the project independently, and the data they collected differed in some important ways, including the timeframe over which citations are considered. Christensen, Dafoe, and Miguel always intended to analyze citations accumulated at a single point in time. Though this was not explicitly stated in our (admittedly too-brief) pre-analysis plan, the regression functional form we proposed in that 2015 document called for controlling flexibly for time since publication, which is important for data collected at one point in time (for articles published in varying years) but would be superfluous if one uses a flow measure of citations.

In summary, it is not totally clear to the reviewer, whether the 2nd part of the study followed the pre-analysis plan and if not, where it deviates. This may have influence on the interpretation of statistical significance (alpha error).

As noted above, we now clarify at various points of the revised paper where the analysis goes beyond the pre-analysis plan, in the interest of transparency. We hope that this helps address the referee’s concern.

Regression model and bias

Citations in year 1 through 5 after article publication do not increase in the treatment journals (AER, AJP); see figure 2, panel E, F). If the cumulative citation advantage (Nov. 2017) is considered, the 2 journals enjoy approximately 40% higher total citations (see figure 2, panel C,D).

a) In figure 2, panel D, the 95% CI is extremely small for 2014 and AJPS Cumulative Citation Advantage (Nov 2017). Apart from this value, the AJPS cumulative citation advantage seems to be small. The authors should explain this.

Thank you for bringing up this issue. The explanation has to do with the amount of variation in total citations in the AJPS articles by 2017, given that they were published only in 2014. As the overall mean and standard deviation of this outcome measure are small overall (as there has been less time for articles to accumulate citations), this leads to tighter confidence intervals. (In the limit, if we had examined citations a mere month, say, after publication of articles, few would have any citations and the confidence interval would be even smaller.) We hope that this provides a logical explanation for the pattern you observed.

b) The apparent trend in the pre-treatment period in economics is a potential concern for the authors and explained by a highly cited AER paper. The reviewer does not understand this argument because AER – QJE is negative in the pre-treatment period.

This has to do with the observation for papers published in 2001, which includes one extremely cited AER paper. As you can see, in 2001 the AER – QJE difference appears to be much more positive relative to the difference in 2002 and 2003, say. Had it not been for the outlier paper published in 2001 in the AER, the AER – QJE difference would have been more stably negative in the pre-period (at least during 2001 through 2003) than it currently appears to be. Of course, there is some further unexplained variation, including the apparent rise in AER citations for papers published in 2004 (compared to 2002 and 2003), which we admittedly cannot fully account for, although some of the year-to-year variation is also presumably being driven by random sampling variation, given the relatively small number of articles published each year in a journal.

c) From the 2-stage least squares regression it is concluded that published articles with posted data enjoyed an increase of 90 to 125 citations over a mean of approximately 100, suggesting nearly a doubling of citations. Here the question is how good the goodness-of-fit of the model is. The R-squared values are not very impressive. Residual plots to determine whether the coefficient estimates are biased could not be found in the supplementary material. The authors should demonstrate the adequacy of the model fit.

Thank you for raising this issue. We have two responses. Our first is that we generated the goodness-of-fit residual plot figure that you propose, and it is included here below. As you can see, and as you guessed, the overall goodness-of-fit is not very high for this model. This figure also appears in the supplementary material as Figure B6.

Our second response is that this approach is not particularly meaningful for the main instrumental variable (IV) two-stage least squares method that we employ in this analysis. It has been shown in econometrics that standard goodness-of-fit measures, like the R-squared and associated metrics, are not meaningful for assessing goodness of fit in IV models. For a well-known discussion, please see: Pesaran, M., & Smith, R. (1994). “A Generalized R^2 Criterion for Regression Models Estimated by the Instrumental Variables Method.” Econometrica, 62(3), 705-710. doi:10.2307/2951666. More broadly, in our view model fit is not central to the exercise we are carrying out: our goal is to obtain unbiased estimates of a particular coefficient, not to create a highly predictive overall model. Goodness-of-fit would matter more for scholars with a different goal than what motivates our paper.

Reviewer #2: Dear authors,

below you can find comments that might improve your manuscript in different perspectives. Please enface my comments as a way to improve your piece of research and I will be more than happy to re-appraise it in a more suitable version.

a) This is a methodological paper and reporting guidelines for methodological research is under discussion. However, you may borrow some items (or even in an intuitive manner) from reporting guidelines from other designs to better guide your writing. Your manuscript in the current form is poorly written and unidentifiable, what is even more crucial for a paper that deals with a question related, at least in some level, to science reproducibility. For example, a simple item like the title does is flawed - it does not indicate if is it an original experiment, a letter, an editorial etc. Standards of reproducible reporting should be addressed (please check www.equator-network.org) and some items might be included accordingly to allow the reproducibility of your paper - title, structured abstract, primary and secondary outcomes, eligibility criteria, search strategy (or the deep description of the approach to address conveniently the journals) and etc. 

Thank you for this important suggestion. As background on the original submission, reporting guidelines are rarely used in our home disciplines in the social sciences (namely, economics, political science, psychology) but we agree that the proposed re-organization will add value and make the article more accessible to scholars across fields, which is our intention. As a result, in the revised version of the paper we have changed the title (along the lines you suggest), have reformatted the paper, and added a flow diagram and text explaining sample determination, per the STROBE guidelines (which appear relevant, if not perfect, for our observational analysis; more information on the STROBE guidelines can be found here: http://www.equator-network.org/reporting-guidelines/strobe/). As you can see, the revised title is: “A Study of the Impact of Data Sharing on Article Citations using Journal Policies as a Natural Experiment”.

You've also waste too much words with non-informative sentences. Please go straight to the point. Avoid also the use of rhetorical sentences and the excess of latin expressions (although suitable at some points).

Thank you for your guidance. We have substantially reduced the word count in the Introduction. The Latin phrases (three in total) are famous quotes and common legal phrases that we feel are the most direct way to express our point; if you see alternatives to some or all of them, we are very open to considering them. If there are other specific wasteful or non-informative sentences that you think remain in the revised version, we would be delighted to modify or cut them in the interest of making the article more direct and parsimonious.

b) Structure your manuscript. Methods, Results and Discussion are completely unorganized. I don't care about the inversion of results and discussion in terms of the length (so, the fact of your discussion was tiny was not a problem for me), if the journal permits. However, they should be informative and address the point to why they were created originally - and you've left a lot of discussion informations in the results section. As for the methods part, this is critical. I must confess I went lost several times when reading your methods due to the way you presented this. It does not favour you neither the reader. Please think careful about it. We need to acknowledge sharply the waste in research nowadays and I don't think your research is waste.

We have sought to clearly demarcate the paper into the four traditional sections, consistent with the reviewer’s requests. This includes major changes in the Methods and Results sections, to make sure that methodological issues are focused in the appropriate section. We have also moved some additional text from Results to Discussion, where this seems appropriate. If our changes are insufficient, we remain open to doing more to improve the structure of the paper, and in particular to make the Methods section clearer for the reader.

c) In your methods, I'm very concerned in the way you've designed your experiment. Propensity scores may balance non-randomized studies at baseline, however, caveats are widely known. Beyond the fact the description of your control group and your propensity score model (I don't know for what variables you've adjusted) are poorly written, I'm hesitant if it is the best method to match groups for your experiments, if is it possible. Given the data on citations related to data-sharing is very naive, I don't know if I would rather prefer to first describe the rate of citations instead of making a comparison. Also, the rate of citation for papers, doesn't matter the data-sharing regimen, is usually low within the first years when there is a salient amount of citations; and, if linear, we would have a decent number only after years. Therefore, you may incur in a comparison with a low frequency of events (with or without inferential analysis) and thus I re-affirm I'm not convinced this is the time to compare citations, but yes for a description. In a future, I would say a comparison within the same journal might be more appropriate, once one will always need to deal with confounding factors that would be difficult to input in a model like the merit of a paper (which makes it more cited), publication bias, gender bias, geographic bias and others. As a last mention, please be very detailed with your control group. I couldn't follow it.

Thank you for pushing us to clarify our approach. We have added the details you requested. We have also added arguments and citations (the well-known Rosenbaum and Rubin 1983 and Dehejia and Wahba 2002 articles) to justify our use of propensity score matching to identify a control group of journals. One important issue to keep in mind is that the statistical analysis depends crucially on the existence of data over time for both the treated and comparison groups of journals; unlike most propensity score analyses, which simply rely on analysis at a single time point (in cross-sectional analysis), our analysis uses propensity score to match journals but then relies on the longitudinal dimension of the data for estimation, along the lines of a difference-in-differences approach. The use of data over time for the same journals is, we believe, an important strength of the analysis since addresses many common concerns associated with propensity score matching.

We regard as appropriate the reviewer’s questions about the right time to measure the effect we examine. We debated this issue prior to formulating our pre-analysis plan. Given the difficulties of examining all possible time periods, we formulated our research plans balancing importance and feasibility. We worry that any time period we could have chosen would have been vulnerable to questions about whether it was the right one. While it is possible our results would have been different had we chosen to examine a different time period, we did not choose our approach endogenously—that is, after having seen the results – and this should help address some concerns about the validity of the findings. We also note that Christensen, Dafoe, and Miguel’s analysis, focusing on citations at a single point in time (in 2017), does help to address some of these concerns about the timeframe of the analysis.

d) The rationale seems weak to me. Do you really think the perception of integrity might improve citations, at least nowadays? I mention the "thinking" because we don't have empirical evidence to support this. I know for a given topic when still under investigated, we have little evidence to support our kick-offs, but I don't know if is it the critical point, or even if data-sharing might incur in the augment of citations (unless for the use of the data, as you acknowledged in your paper). Citations are a multidimensional approach and a multivariable-explained phenomenon. Anecdotally, I'm not convinced data-sharing may play a role in citations within the myriad of variables - if researchers nowadays didn't perceive the value of data-sharing in various scenarios, I would say that an association is still harder (although not implausible by nature). Even, as a data-sharing researcher, I think data sharing should be observed as a way to reduce the waste in research; to improve the reproducibility and verifiability of a given finding; to augment the utility of a finding; and an ethical obligation of every single researcher regardless of the experimental addressed designed - what you brilliantly did. In time, congratulations by the way you shared the data. I didn't re-analyze your findings, but it is clear your compliance with reproducibility standards in that way.

Thank you for this comment, which is useful. You are correct that a journal’s or an article’s impact is determined by many more things than just whether the data happen to be shared. Even though this is one among many different factors, we see it as increasingly important in the era of open science, and designed our study to detect its influence. Moreover, there is an influential body of research (which we cite in our article) arguing that it is exactly the case that articles that share data are cited more often; however, we believe that this existing research has many methodological limitations that we are able to address in the current paper. 

Nevertheless, we recognize the limitations of the evidence regarding how sharing data might affect perceptions of integrity, and have greatly reduced the discussion of whether sharing data should result in improved researcher reputation.

---

## [Decision Letter · Decision Letter 1]

2 Oct 2019

PONE-D-19-18102R1

A study of the impact of data sharing on article citations using journal policies as a natural experiment

PLOS ONE

Dear Dr. Christensen,

Thank you for submitting your manuscript to PLOS ONE. After careful consideration, we feel that it has merit but does not fully meet PLOS ONE’s publication criteria as it currently stands. Therefore, we invite you to submit a revised version of the manuscript that addresses the points raised during the review process.

The first referee found the manuscript improved (I agree) and he made two important comments. Please follow these comments.

In addition, in your abstract, please provide explicitly more details about your methods and results, expressed in a quantitative way using point estimates and 95 % confidence interval for your main outcomes. Please also make a short statement of your main limitations (see below) in the introduction. 

The second reviewer still thinks that the reporting must be improved. And I still agree with him because this is a core part of reproducible research.

To be more concrete here are some important suggestions for improvement : 

- In the method section, please provide all the details necessary to reproduce your selection of journals. Please try to adapt the PICOS formulation of the PRISMA statement for this. Please provide more details concerning the matching of journals explaining specifically why 17 journals were matched with 13 journals (this should be also addressed in the discussion section). Please make a specific paragraph "changes to the initial protocol". Please use subheadings to differentiate the different parts of the project (e.g. broad analysis / deep analysis / and a specific part statistical analysis). Please do not use the word "causal" in the manuscript, as you are only exploring association and not causal relationships : e.g. when you state : "equation 2 uses predicted data availability to estimate the causal...") ;

- In the result section, please provide all quantitative results of the model described in the method section. Importantly, I would expect your to report point estimates and 95 % CI. Table(s), or results written in the text must appear in the paper. As it is now, there are only qualitative descriptions/informations about the results. This section must report data, be factual and should not present a discussion of the results. Please move any interpretation to the discussion section. 

- In the discussion section, please start with a description of the principal finding and here please interpret your findings. Please add an important emphasis on risk of bias and especially confounding. Please make it very clear that this study is about associations and not causal relationships. Please also discuss the rational and the previous literature about the expected association between citations and data sharing practices. Please try to be systematic and exhaustive in your appraisal of the literature.

In my opinion, the manuscript therefore still needs major revision, on the form, before formal acceptance. 

We would appreciate receiving your revised manuscript by Nov 16 2019 11:59PM. To enhance the reproducibility of your results, we recommend that if applicable you deposit your laboratory protocols in protocols.io, where a protocol can be assigned its own identifier (DOI) such that it can be cited independently in the future. For instructions see: http://journals.plos.org/plosone/s/submission-guidelines#loc-laboratory-protocols

We look forward to receiving your revised manuscript.

Kind regards,

Florian Naudet, M.D., M.P.H., Ph.D.

Academic Editor

PLOS ONE

Additional Editor Comments (if provided):

None

Reviewers' comments:

Reviewer's Responses to Questions

**Comments to the Author**

1. If the authors have adequately addressed your comments raised in a previous round of review and you feel that this manuscript is now acceptable for publication, you may indicate that here to bypass the “Comments to the Author” section, enter your conflict of interest statement in the “Confidential to Editor” section, and submit your "Accept" recommendation.

Reviewer #1: All comments have been addressed

Reviewer #2: All comments have been addressed

2. Is the manuscript technically sound, and do the data support the conclusions?

Reviewer #1: Yes

Reviewer #2: Partly

3. Has the statistical analysis been performed appropriately and rigorously? 

Reviewer #1: Yes

Reviewer #2: Yes

4. Have the authors made all data underlying the findings in their manuscript fully available?

Reviewer #1: Yes

Reviewer #2: Yes

5. Is the manuscript presented in an intelligible fashion and written in standard English?

Reviewer #1: Yes

Reviewer #2: Yes

6. Review Comments to the Author

Reviewer #1: The comments of the reviewers were adressed and the manuscript has been considerably improved. There are two minor issues, which need to be adressed:

a) The abstract should be structured (introduction, methods, results, ...)

b) The authors have responded to the question about goodness of fit of 2-stage least squares regression in the "response to reviewers".In the manuscript, This point is not really tackled in the manuscript (only a comment that the first stage in the two-stage least squares regression is strong). The authors should include a statement concerning potential bias of estimates of coefficients in the text (along the lines of their argumentation in the "response to reviewers").

Reviewer #2: Dear authors,

thank your for addressing the points and checking my comments. They are partially addressed and I have more several comments on the format, however, I don't they are worthy now given the timing - you will need to re-write the manuscript, what I don't want. So, for now, I'm only on the merit of the content. Nothing more to say, please add/expand your limitation paragraph for future directions and the ICMJE Statement for Data-Sharing (your conditions).

7. PLOS authors have the option to publish the peer review history of their article (what does this mean?). If published, this will include your full peer review and any attached files.

Reviewer #1: No

Reviewer #2: Yes: Lucas Helal

---

## [Author Response · Author response to Decision Letter 1]

11 Nov 2019

November 3, 2019

Response to Reviewers

Thank you to our reviewers for their continued careful review of our paper. We respond point by point to the comments that were highlighted by the editor. 

In addition, in your abstract, please provide explicitly more details about your methods and results, expressed in a quantitative way using point estimates and 95 % confidence interval for your main outcomes. 

Thank you for this suggestion. We now specifically mention both of our main methods (matching, instrumental variables) and describe the results quantitatively, reporting the standard error of our main estimate, as is reported in the tables and as is customary in our disciplines of economics and political science. 

Please also make a short statement of your main limitations (see below) in the introduction. 

The last sentences of the introduction now discuss our generalizability and limitations (namely, that we cannot address why articles are cited more, and we use observational not experimental data.)

- In the method section, please provide all the details necessary to reproduce your selection of journals. Please try to adapt the PICOS formulation of the PRISMA statement for this. Please provide more details concerning the matching of journals explaining specifically why 17 journals were matched with 13 journals (this should be also addressed in the discussion section).

We now clearly address this in the methods section with the following language:

“To account for the possibility of events that influenced both the change in journal policy and citation rates that could bias estimates, we collected comparable data for two natural comparison sets. We gathered data on theoretical articles published in the same journals; since these do not use empirical data, their citations should be largely unaffected by any change in data posting policy. We also matched the 17 treatment journals (which began to require data sharing) to control journals (which did not), and collected comparable citation data for empirical articles published in these control journals. 

Our control journals are derived using conventional one-to-one propensity score matching with replacement from top-ranked journals that most closely match the treatment journals on six SCImago criteria (8,9). Our objective was to identify control journals that did not require data posting, but that were otherwise as similar as possible to the treatment journals in terms of observable characteristics. Accordingly, we began with non-treatment journals -- that never required data posting – from the same SCImago “Top 250” list of journals where we searched for our treatment journals. We matched treatment to control journals using the six indicators used to create the SCImago list itself. These criteria vary by journal and are posted by Scimago on its website. The six variables are: a) the journal’s h-index; b) the total number of citable documents published in the journal’s last three years; c) citations per document over the last two years; d) references per document; e) the country where the journal was published; and f) the category of the journal. Since there are only two countries where our treatment journals were published, we created a binary variable for journals published in the UK, leaving US journals as the default. And since we only have a limited number of treatment journals, we consolidated journal category into eight areas: a) Biology; b) Ecology; c) Economics; d) Medicine; e) Molecular Biology; f) Multidisciplinary; g) Sociology and Political Science; and h) Miscellaneous. 

After creating the data set, we created a binary variable, with unity for our treatment journals and zero for all remaining journals (potential control journals). We then estimated a cross-sectional probit equation; the results are tabulated in our online supplement. After estimating the probit model, we then matched each of the treatment journals to a single control journal, using the closest possible journal, by predicted probit score within journal category. Sometimes this resulted in more than one treatment journal being matched to the same control journal. Both the control journals and the probit regression estimates themselves are freely available online. We are left with a set of 13 unique control journals along with 17 treatment journals; thus the MR analysis included data from a total of 30 distinct scientific journals. Appendix A provides the list of journals and more details on the data construction procedure.”

Please make a specific paragraph "changes to the initial protocol". 

This information was already explicitly included in the appendix, but we now include it in the main methods section as well, as follows:

“In 2015, the CDM team pre-registered their analysis (https://osf.io/hv97m/). We deviate from it here in the following ways: First, we did not think to exclude articles that do not use data. (Results on the full sample, described in Appendix Tables B21 to B23, are generally weaker.) We also planned to control for time using cubic functions of months since publication. The results presented in the paper instead use year-discipline fixed effects to more flexibly control for time, as can be seen in the equations below. (Models with the cubic function yield nearly identical results.) In September 2017 CDM, expanded their analysis from the pre-registered field of political science (from which they had seen results) to include the economics portion of the project; this was before seeing any results from the broad analysis conducted by DM.” 

Please use subheadings to differentiate the different parts of the project (e.g. broad analysis / deep analysis / and a specific part statistical analysis). 

This is a good suggestion that will help make our paper clearer. We have added subsections to both methods and results for both the broad analysis and deep analysis. As both of these analyses are clearly statistical in nature, we do not specifically label either as statistical.

Please do not use the word "causal" in the manuscript, as you are only exploring association and not causal relationships : e.g. when you state : "equation 2 uses predicted data availability to estimate the causal...") ;

This was the only instance of the word “causal” in our paper. We are happy to modify the language here to say that “equation 2 uses predicted data availability to attempt to estimate the impact on number of total citations under assumptions we discuss below,” which we believe is completely accurate. We have also added citations to the 1996 Angrist, Imbens, and Rubin JASA paper “Identification of Causal Effects Using Instrumental Variables” (which has been cited over 5,000 times) and the 1994 Imbens and Angrist Econometrica paper “Identification and Estimation of Local Average Treatment Effects” (cited over 4,400 times) which both strongly make the case that instrumental variables can be used to estimate causal effects under certain assumptions. This is in addition to the Dehijia and Wahba (2002) REStat paper with 4,600 citations and the Rosenbaum and Rubin (1985) paper with 23,800 citations that discuss propensity score matching and causality, both of which we already cited. We discuss and test these assumptions in our paper to the extent possible, and are forthright about the evidence that we may violate one of these assumptions. We thus feel strongly that it is appropriate to state that we attempted to estimate causal effects, because that is, in fact, what we attempted to do.

- In the result section, please provide all quantitative results of the model described in the method section. 

Importantly, I would expect you to report point estimates and 95 % CI. Table(s), or results written in the text must appear in the paper. As it is now, there are only qualitative descriptions/informations about the results. This section must report data, be factual and should not present a discussion of the results. Please move any interpretation to the discussion section. 

Thank you for this suggestion. We have checked that every numerical result in the paper is either:

(a) directly from a table that appears in the paper itself, if a main result, with standard errors included 

(b) from a discussion of a figure, and thus not an exact statistical result, and thus not possessing a standard error

(c) from a table that appears in the appendix, if not a main result and instead the result of a robustness check, with the exact appendix table number specified. 

We have moved additional results tables into the main paper, but in order to be transparent and exhaustive, the appendix includes 40 tables, and certainly not all of these belong in the main paper. 

- In the discussion section, please start with a description of the principal finding and here please interpret your findings. Please add an important emphasis on risk of bias and especially confounding. Please make it very clear that this study is about associations and not causal relationships. Please also discuss the rational and the previous literature about the expected association between citations and data sharing practices. Please try to be systematic and exhaustive in your appraisal of the literature.

Thank you for these suggestions. We now begin the discussion section with a discussion of our main results. We discuss bias in both of the methods sections, and have lengthened the discussion regarding both of the main methods, their assumptions, and potential biases.

Though this article is not intended as a review piece, an updated search did provide a few extra citations, which we discuss in more detail. If you are aware of others that we have not identified, we are open to suggestions. Regarding the interpretation of our results, and in particular whether we are estimating causal impacts, please refer to our discussion above of your earlier point.

Sincerely,

Garret Christensen, Allan Dafoe, Edward Miguel, Don A. Moore, and Andrew K. Rose

---

## [Editor Report · Decision Letter 2]

15 Nov 2019

A study of the impact of data sharing on article citations using journal policies as a natural experiment

PONE-D-19-18102R2

Dear Dr. Christensen,

We are pleased to inform you that your manuscript has been judged scientifically suitable for publication and will be formally accepted for publication once it complies with all outstanding technical requirements. Thank you for all your edits and thanks to the two reviewers for their careful reading and comments. 

With kind regards,

Florian Naudet, M.D., M.P.H., Ph.D.

Academic Editor

PLOS ONE
---

## [Editor Report · Acceptance letter]

21 Nov 2019

PONE-D-19-18102R2 

A study of the impact of data sharing on article citations using journal policies as a natural experiment 

Dear Dr. Christensen:

I am pleased to inform you that your manuscript has been deemed suitable for publication in PLOS ONE. Congratulations! Your manuscript is now with our production department. 

With kind regards,

on behalf of

Dr. Florian Naudet 

Academic Editor

PLOS ONE